# Agricultural Policy Environmental eXtender (APEX) Simulation of Spring Peanut Management in the North China Plain

**Jie Zhao [1,2,†], Qingquan Chu [1,†], Mengjie Shang [1], Manyowa N. Meki [2], Nicole Norelli [2], Yao Jiang [1], Yadong Yang [1], Huadong Zang [1], Zhaohai Zeng [1,\*] and Jaehak Jeong [2,\*]**

[1] College of Agronomy and Biotechnology, China Agricultural University, No. 2 Yuanmingyuan West Road, Beijing 100193, China

[2] Blackland Research Center, Texas A&M AgriLife Research, Texas A&M University, 720 East Blackland Road, Temple, TX 76502, USA

[\*] Correspondence: zengzhaohai@cau.edu.cn (Z.Z.); jeongj@tamu.edu (J.J.); Tel.: +86-10-6273-1211 (Z.Z.); +1-254-774-6118 (J.J.)

[†] These authors contributed equally to this work.

**Abstract:** Spring peanut is a valuable alternative crop to mitigate water scarcity caused by excessive water use in conventional cropping systems in the North China Plain (NCP). In the present study, we evaluated the capability of the Agricultural Policy Environmental eXtender (APEX) model to predict spring peanut response to sowing dates and seeding rates in order to optimize sowing dates, seeding rates, and irrigation regimes. Data used for calibration and validation of the model included leaf area index (LAI), aboveground biomass (ABIOM), and pod yield data collected from a field experiment of nine sowing dates and seeding rate combinations conducted from 2017 to 2018. The calibrated model was then used to simulate peanut yield responses to extended sowing dates (5 April to 4 June with a 5-day interval) and seeding rates (15 plants m$^{-2}$ to 50 plants m$^{-2}$ with a 5 plants m$^{-2}$ interval) using 38 years of weather data as well as yield, evapotranspiration (ET), and water stress days under different irrigation regimes (rainfed, one irrigation before planting (60 mm) or at flowering (60 mm), and two irrigation with one time before planting and one time at flowering (60 mm each time) or at pod set (60 mm each time)). Results show that the model satisfactorily simulates pod yield of peanut based on $R^2$ = 0.70, index of agreement (*d* value) being 0.80 and percent bias (PBIAS) values ≤4%. Moreover, the model performed reasonably well in predicting the emergence, LAI and ABIOM, with a $R^2$ = 0.86, *d* = 0.95 and PBIAS = 8% for LAI and $R^2$ = 0.90, *d* = 0.97 and PBIAS = 1% for ABIOM, respectively. Simulation results indicate that the best combination of sowing dates and seeding rates is a density of 35–40 plants m$^{-2}$ and dates during early-May to mid-May due to the influence of local climate and canopy structure to the growth and yield of peanut. Under the optimal sowing date and plant density, an irrigation depth of 60 mm during flowering gave a pod yield (5.6 t ha$^{-1}$) and ET (464 mm), which resulted in the highest water use efficiency (12.1 kg ha$^{-1}$ mm$^{-1}$). The APEX model is capable of assessing the effects of management practices on the growth and yield of peanut. Sowing 35–40 plants m$^{-2}$ during early-May to mid-May with 60 mm irrigation depth is the recommended agronomic practice for peanut production in the water-constrained NCP.

**Keywords:** APEX model; alternative crop; peanut; sowing date; seeding rate; water use efficiency

## 1. Introduction

The increasing need for high-quality food with the minimum environmental impact has led to renewed interest in grain legumes to maintain sustainable crop production [1–3]. Peanut (*Arachis hypogaea* L.) plays a critical role in oil security as well as being an important cash crop in China. China is the largest producer of peanut in the world [4]. The North China Plain (NCP), which is often referred to as the breadbasket of China, provides more than 72% of the nation's wheat and 33% of its maize production [5]. Such achievements have heavily depended on continuous overexploitation of groundwater for irrigation and excessive mineral nitrogen (N) fertilizer application to meet the water and N requirement of intensive wheat–maize cropping systems [6,7]. Adjustment and optimization of cropping systems could be an effective strategy to mitigate groundwater depletion and reduce N input to ensure sustainable food production [8–11]. Alternative cropping systems include spring maize monoculture [12,13], a 2-year system of winter wheat/summer maize–spring maize [11] or double maize cropping systems [14] to allow replenishment of soil water reserves and fertility. Peanut has significant beneficial effects on subsequent crops in rotation with its inherent capacity for symbiotic atmospheric nitrogen fixation and considerable economic return, making it a valuable alternative crop in cropping structure adjustment [15,16].

Under the policy of "fallow in winter wheat season and rainfed in rainy season", which aimed to reduce the winter wheat planting area [12,17], the cultivated area of spring peanut (or early sowing peanut, planting during early April to late May) increased steadily. Contrast to the fixed sowing dates (5 June to 20 June, planting after winter wheat) of summer peanut, spring peanut can be planted over a considerable part of the season under suitable conditions in the NCP. However, the choosing of a sowing date must have consideration of alleviating drought or high-temperature stress during critical stages of peanut growth [18]. Sowing dates for peanut have been studied extensively in most peanut-growing countries [19–22]. These studies indicate that matching the phenology of the peanut crop to the duration of favorable conditions by selecting the most suitable sowing dates to avoid periods of stress is crucial for maximizing peanut yield. Moreover, appropriate planting density is needed to fully explore the peanut's genetic potential for better yield. Planting density not only determines competition for light, water, and nutrients, but also regulates vegetative growth and reproductive development and ultimately controls the distribution of dry matter between the organs and pod yield [23–25]. Peanut seed is also more expensive than the seed of most row crops, making it important to reduce seed requirement and costs. In addition, supplementary irrigation during the critical period of water requirement is crucial to increase the productivity of peanut [26,27]. Despite these resources, research on optimum sowing dates, seeding rates, and irrigation regimes of spring peanut is limited particularly in the NCP.

Experimental approaches aimed at optimizing management practices are costly and time-consuming because of the necessity in capturing the potential impact of long-term climate. Opportunities exist to examine the effect of management practices through process-based crop models. The Agricultural Policy/Environmental eXtender (APEX) is a process-based model for managing landscapes, whole farms, and small watersheds to achieve sustainable production and maintain environmental quality [28]. APEX is capable of simulating crop growth, hydrology, soil erosion, soil carbon sequestration, nutrient cycling, and losses [29]. Due to its wide applicability and strong capability, the APEX model has been extensively used to simulate grain yield, irrigation regimes, carbon, and nitrogen management under a variety of management conditions across the world. Wang et al. (2008) successfully predicted maize grain yield and soil organic carbon in an erosion-prone area in Iowa, US [30]. Cavero et al. (2012) assessed irrigation and N management strategies using APEX in three Mediterranean watersheds [31]. Meki et al. (2011; 2013) simulated maize residue removal effects on nutrient, soil organic carbon losses, and subsequent soybean yield and nitrogen dynamics in the Upper Mississippi River Basin [32,33]. Soybean yield potential, yield gap under rainfed conditions, and crop-water production functions were identified in a humid region of Mississippi, USA [34–36]. Luo and Wang (2019) evaluated corn, rice, and wheat grain yields and sediments under various irrigation scenarios and tillage methods in southwest

China [37]. Plotkin et al. (2013) used the APEX model to simulate peanut yield, hydrology, and pesticide losses for a peanut/cotton rotation under the conditions of the Southern Atlantic Coastal Plain [38]. Van Liew et al. (2017) evaluated crop yields and sediments under peanuts and cotton dominated cropping systems on the Little River watersheds in Georgia [39]. Overall, these studies indicate that the APEX model is capable of successfully modeling the impacts of climate, soil, cropping system, tillage, irrigation, and management practices on crop growth and productivity.

Although simulation models have the potential for optimizing crop management, they must be calibrated and validated before they can be applied in the NCP. We conducted a field experiment to compare peanut growth and yield response to seasons, sowing dates, and seeding rates. Data from this experiment was used to calibrate and validate the APEX model. The objectives of this study are to (1) evaluate the performance of APEX on estimating peanut growth and crop yield, (2) assess peanut yield under extended scenarios of sowing dates and seeding rates, and (3) identify the best practice for maximizing water productivity under different irrigation regime scenarios. It is of great significance to improve peanut water productivity and further the application of the APEX model under water constrained areas in the NCP.

## 2. Materials and Methods

### 2.1. Experimental Site Description

The field experiment was conducted at the Wuqiao Experimental Station of the China Agricultural University in Wuqiao County (37°37′ N, 116°26′ E, altitude 18–21 m) of Hebei Province, China (Figure 1). The climate of the study area is a warm temperate and continental monsoon climate, typically with hot, rainy summers and dry, cold winters. Monthly trends in rainfall and average air temperature during the last 38 years including 2017 and 2018, the experiment years, are shown in Figure 2. The annual mean air temperature over the last 38 years was 12.9 °C, the annual accumulated temperature ($\geq 0$ °C) was 4826 °C, the frost-free period was 201 days, and the annual number of sunshine hours was 2724 h. The average annual precipitation over the last 38 years was 554 mm, 70% of which occurred from June to August (Figure 2). The predominant soil is classified as Calcaric Fluvisol with a silt loam texture. Table 1 lists the soil texture and hydraulic properties for each layer of the 1.2 m profile. The station is representative of the overall agricultural production and climate conditions of the NCP. The winter wheat–summer maize rotation is the main cropping system and irrigated cropland accounts for about 70% of the total land in this area. Groundwater is the major source for irrigation as almost no surface water is available. The groundwater table is approximately 17 m below ground surface.

**Table 1.** Soil texture and hydraulic properties of the soil profile at the Wuqiao experimental site.

| Soil Layer (m) | BD (g cm$^{-3}$) | Particle Fraction (%) | | | Texture | SATC (mm h$^{-1}$) | UW (m m$^{-1}$) | FC (m m$^{-1}$) | pH |
|---|---|---|---|---|---|---|---|---|---|
| | | Sand | Silt | Clay | | | | | |
| 0–0.2 | 1.33 | 16.2 | 69.2 | 14.6 | Silt Loam | 4.4 | 0.10 | 0.30 | 8.1 |
| 0.2–0.4 | 1.52 | 18.0 | 67.7 | 14.4 | Silt Loam | 1.1 | 0.12 | 0.27 | 8.3 |
| 0.4–0.6 | 1.46 | 12.1 | 82.1 | 5.8 | Silt Loam | 5.4 | 0.12 | 0.27 | 8.3 |
| 0.6–0.8 | 1.45 | 8.9 | 75.9 | 15.2 | Silt Loam | 7.9 | 0.12 | 0.29 | 8.2 |
| 0.8–1.0 | 1.46 | 4.7 | 81.0 | 14.3 | Silt Loam | 7.1 | 0.13 | 0.31 | 8.3 |
| 1.0–1.2 | 1.41 | 2.6 | 86.3 | 11.1 | Silt | 8.3 | 0.18 | 0.32 | 8.6 |

Note: BD is bulk density, SATC is soil saturated hydraulic conductivity, UW is the wilting point, FC is field capacity.

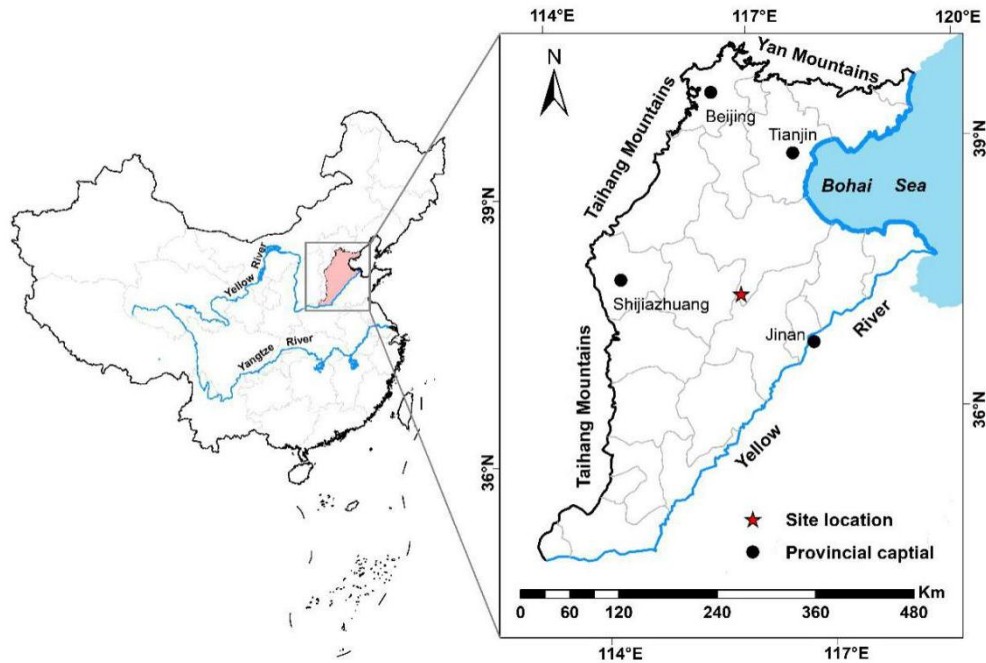

**Figure 1.** Location of Wuqiao County and the North China Plain.

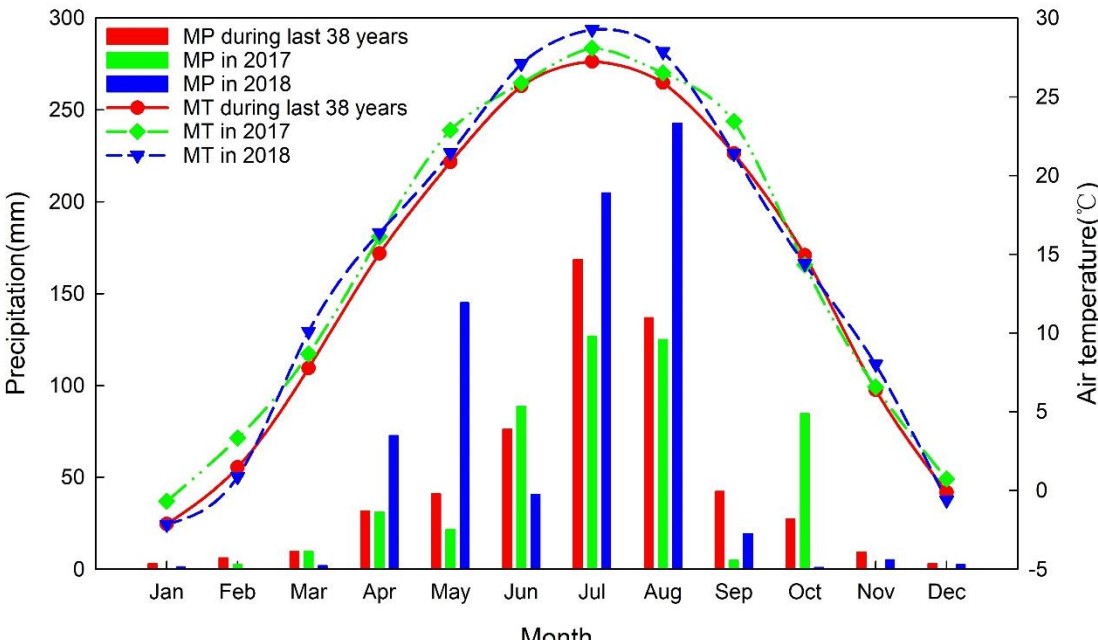

**Figure 2.** Monthly precipitation and air temperature (MP = mean precipitation, MT = mean air temperature) during 2017, 2018, and the last 38 years (1981–2018) at Wuqiao County, Hebei Province, China.

## 2.2. Experimental Design and Data Collection

Measured data for evaluating the APEX model's performance in simulating peanut growth and yield were obtained from an experiment conducted during the growing seasons of 2017 and 2018. A split-plot design with three replicates was used each year for data collection. The main plot had three sowing dates, i.e., late April (25 April, T1), early May (5 May, T2), and mid-May (15 May, T3); and sub-plots had three levels of seeding rates, 24 (D1), 30 (D2), and 36 (D3) plants m$^{-2}$. Individual sub-plots were built with 7.5 cm high, 40 cm wide peanut ridging beds with two rows, the inner rows were spaced 30 cm apart and outer rows were spaced 81 cm for D1, 59 cm for D2, and 44 cm for D3,

respectively (Figure 3). Plant spacing was 15 cm for all treatments and each plot was 5 m by 7 m in size. Each plot had 6, 7, and 9 raised beds for D1, D2, and D3, respectively. The rows were oriented north–south. Yuhua 25, a high yield median-large size peanut cultivar, planted with two seeds per hill, was selected from among 9 peanut varieties used in a previous experiment [40]. Flood irrigation (60 mm, recorded with totalizing water meters) was applied to all treatments to facilitate seed germination. All plots were fertilized prior to planting with 171 kg N ha$^{-1}$ in the form of urea (225 kg ha$^{-1}$, 46% N) and diammonium phosphate (375 kg ha$^{-1}$, 18% N), 173 kg P$_2$O$_5$ ha$^{-1}$ in the form of diammonium phosphate (375 kg ha$^{-1}$, 46% P$_2$O$_5$), and 150 kg K$_2$O ha$^{-1}$ in the form of potassium sulfate (300 kg ha$^{-1}$, 50% K$_2$O). The herbicide Pendimethalin was applied at a rate of 0.99 kg ha$^{-1}$ after sowing. The insecticides Chlorantraniliprole and *Bacillus thuringiensis* were applied as needed at a rate of 0.03 kg ha$^{-1}$ at pegging stage and 600 mL ha$^{-1}$ (8 IU mL$^{-1}$) at early podding stage to control grubs (*Phyllophaga* spp.) and aphids (Aphidoidea), respectively. The fungicides Carbendazim and Mancozeb were applied both at an early podding stage at a rate of 0.68 kg ha$^{-1}$ and 1.13 kg ha$^{-1}$ to control peanut leafspot (*Cercospora arachidicola*), respectively. Although measures were implemented in both experimental years to avoid diseases and pests damage, there was still some plant damage due to the inefficiency of pesticides caused by frequent and heavy rainfall events during the 2018 growing season.

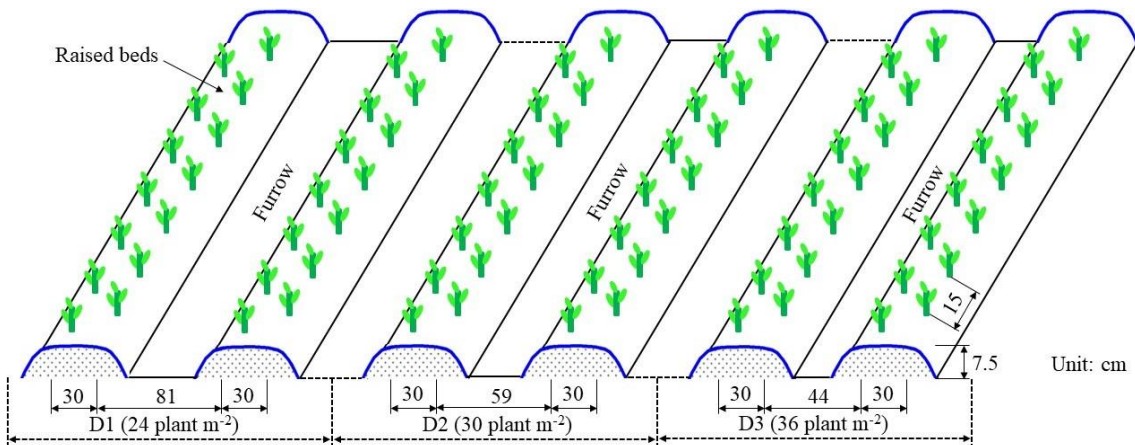

**Figure 3.** Schematic diagram of the arrangement of planting rows and raised beds under the three seeding rates.

Data were collected on plant growth and development. The vegetative and reproductive growth stages including emergence (VE, cotyledons near the soil surface with some part of the plant visible in half of the seedlings.) and harvest maturity (R8, plants with 70%–80% mature pods) were determined using the system developed for peanut by Boote (1982) [41]. These stages were assumed to have occurred when at least 50% of the plants in a plot had reached that stage. Maximum plant height was measured at the R8 growth stage [41]. Growth analysis data were collected four times during the peanut growing season (Table 2). Measured plant variables included dry weight and leaf area index. Five consecutive plants in the central rows of each plot were sampled each time and washed to remove soils attached to the pods. Plant components were separated to determine leaf area and to partition dry matter into plant components such as leaves, stems, pegs, pods, and seeds. After recording the leaf area of each sample with a leaf area meter (Yaxin-1241, Beijing Yaxin Science Instrument Technology Co., Ltd., Beijing, China). The samples were oven-dried at 65 °C for 48 h and weighed to determine dry weight. At each harvest date, the peanut pods were dug up by hand with conventional tools. The commercial yield was assessed by harvesting 3-m lengths of row by hand from the central four rows, and drying until pods reached about 12% moisture content. Then, 500 g of pods as representative subsamples were hand-shelled to determine kernel yields and shelling percentages.



**Table 2.** Agronomic and potential heat unit (PHU) data of peanut in 2017 and 2018.

| Season | Sowing Date | VEo | VEs | Sampling Date | | | | Harvesting Date | Growth Duration | P (mm) | PHU (°C) |
|---|---|---|---|---|---|---|---|---|---|---|---|
| 2017 | 4/25 | 5/7 | 5/8 | 6/3 | 6/23 | 7/26 | 9/1 | 9/2 | 131 | 362 | 2000 |
| | 5/5 | 5/14 | 5/15 | 6/10 | 6/30 | 8/6 | 9/9 | 9/10 | 129 | 355 | 2050 |
| | 5/15 | 5/23 | 5/23 | 6/21 | 7/7 | 8/10 | 9/17 | 9/18 | 126 | 355 | 2030 |
| 2018 | 4/25 | 5/7 | 5/8 | 6/3 | 7/17 | 8/14 | 9/1 | 9/2 | 131 | 634 | 2040 |
| | 5/5 | 5/15 | 5/15 | 6/6 | 7/19 | 8/20 | 9/03 | 9/5 | 124 | 621 | 2040 |
| | 5/15 | 5/22 | 5/23 | 6/13 | 7/23 | 8/20 | 9/8 | 9/9 | 118 | 619 | 2040 |

Note: VEo is observed emergence date, VEs is simulated emergence date, P is growing season precipitation.

### 2.3. APEX Model Description and Input Parameters

APEX 1501 (last updated on 15 January 2019) was used in this study to develop APEX models for the peanut field trials. It has a user-friendly Windows interface, which is designed and maintained by the Blackland Research and Extension Center Modeling Team [28]. APEX was developed to expand EPIC's capabilities to simulate multiple subareas simultaneously [42], which means it can be run for single fields similar to EPIC [43]. APEX uses EPIC's general crop growth model that is capable of simulating various types of over 100 annual and perennial crops [44]. Each crop is defined by 56 crop parameters with unique values. APEX crop parameters and their values used for peanut simulation in this study are listed in Table 3. The APEX model is a continuous model which operates on a daily time step and can simulate up to hundreds of years [30,45]. Crop growth is simulated based on daily heat unit accumulation [44]. Potential increase in biomass is calculated daily based on intercepted photosynthetic active radiation and radiation use efficiency, and is then adjusted to actual growth depending on the level of daily stresses caused by water, nutrients, temperature, and aeration. If the crop is under the full stress condition on a day, the crop does not gain biomass for the day. In APEX, crop yield is estimated by using the harvest index, which is defined as the ratio of economic yield to aboveground biomass at maturity [46]. Water or temperature stress may cause a reduction in harvest index if they occur during sensitive stages such as flowering or grain-filling periods.

**Table 3.** Peanut plant parameters used to calibrate the Agricultural Policy Environmental eXtender (APEX) model. The adjusted parameters are shown in boldface.

| Parameters | Description | Default | Adjusted |
|---|---|---|---|
| WA | Biomass-Energy Ratio | 30.00 | 30.00 |
| HI | Harvest index | 0.40 | 0.40 |
| DMLA | Maximum potential leaf area index | 5.00 | 5.00 |
| DLAI | Fraction of growing season when leaf area declines | 0.85 | 0.75 |
| DLAP1 | First point on optimal leaf area development curve | 15.01 | 12.05 |
| DLAP2 | Second point on optimal leaf area development curve | 50.95 | 50.70 |
| RLAD | Leaf area index decline rate parameter | 1.00 | 0.20 |
| RBMD | Biomass–energy ratio decline rate parameter | 0.50 | 1.00 |
| PPLP1 | Plant Population for Crops and Grass-1st Point on curve | 3.10 | 10.55 |
| PPLP2 | Plant Population for Crops and Grass-2nd Point on curve | 10.90 | 40.95 |

All the agricultural production processes are accounted for in the APEX model in a very detailed fashion [29]. APEX inputs required for model construction include weather data, soil properties, crop data, and management information. APEX simulation is mainly driven by weather data includes daily solar radiation, precipitation, maximum air temperature, and minimum air temperature. In addition, measured daily wind speed and relative humidity were added to the daily weather input to select the Penman or Penman–Monteith method for estimating potential evapotranspiration [46,47]. Weather data were added into the existing APEX weather database by WeatherImport, a stand-alone weather

generator. Soil properties include layer depth, bulk density, field capacity, permanent wilting point, percentage sand, percentage silt, percentage organic carbon, and saturated hydraulic conductivity. Crop data includes growth, yield, leaf area index (LAI), harvest index, root depth, crop height, biomass, and plant population. Crop management inputs include the rate and/or date of plowing, sowing, fertilizer, irrigation, pest control, cultivar, and harvest. We used the harvesting plant density, around 90% of designed density in both years, as the input density in the operation file. As the current version cannot simulate twin rows in one raised bed, one row in one bed was adopted in the model with the average row interval of 0.56, 0.44, and 0.37m for D1, D2, and D3, respectively. The simulated peanut seed yield was adjusted to pod yield based on 5% of water content in seeds and 71% of shelling percentage for uniformity of comparisons.

*2.4. APEX Model Calibration and Validation*

Before a model is applied to simulate various scenarios, model calibration is often needed to obtain reliable results [48]. In this study, APEX models for the peanut trials were calibrated and validated using a field measured pod yield, above-ground plant biomass (ABIOM), and LAI. APEX was calibrated for the 2017 growing season and then validated using data collected in 2018, with a warm-up simulation of 6 years (2011–2016) to stabilize the soil pools (e.g., carbon and nitrogen). Calibration was carried out manually using a trial-and-error method in which sensitive influential parameters and inputs were adjusted within their recommended ranges. Model outputs were analyzed graphically by comparing simulated versus measured LAI, ABIOM, and yield data, in addition to simultaneously assessing model performance statistics [49]. Parameter sensitivity assessed during manual adjustments indicated that peanut biomass and yield were responsive to several input parameters, including crop biomass energy ratio (WA), harvest index (HI), the maximum LAI (DMLA), and the fraction of growing season when LAI starts to decline (DLAI) and, as expected, soil properties, weather, and management practices.

*2.5. Simulation Scenarios*

To optimize seeding rates and sowing dates, APEX scenarios that combine 8 seeding rates (15–50 plants m$^{-2}$, 5 plants m$^{-2}$ interval) and 13 sowing dates (5 April to 4 June, 5 days interval) were developed. In total, 104 scenarios under rainfed or full irrigation conditions were constructed and assessed for 38 years (1981 to 2018). These scenarios were made sufficiently broad to evaluate the effects of irrigation regimes on peanut pod yield and water use efficiency under the best seeding rates (40 plants m$^{-2}$) and sowing dates (10 May). Five irrigation regime scenarios were designed, namely W0 (rainfed), W11 (60 mm depth before planting), W12 (60 mm depth at flowering), W21 (60 mm depth before planting and 60 mm depth at flowering), and W22 (60 mm depth before planting and 60 mm depth at pod set). In addition, in order to explore crop responses to irrigation in different rainfall category years, simulation years (1981 to 2018) were grouped into three precipitation regimes according to the rainfall amount during each growing seasons using an empirical frequency analysis [50,51]: Wet years (25% guaranteed rate of precipitation), normal years (50% guaranteed rate of precipitation), and dry years (75% guaranteed rate of precipitation). Model simulation results were evaluated for yield responses as well as water use efficiency (WUE). WUE (kg ha$^{-1}$ mm$^{-1}$) for the pod yield (kg ha$^{-1}$) was calculated as follows [52]:

$$\text{WUE} = \frac{PY}{ET} \tag{1}$$

where *PY* is the pod yield (kg ha$^{-1}$) and *ET* (mm) was simulated by the APEX model using Penman–Monteith methods. Water stress (WS), defined as actual plant uptake/potential plant water use rate, was used to identify the timing and duration of water deficit under the three precipitation regimes.

### 2.6. Data Analysis and Model Performance Criteria

Data analysis of the field experiments was conducted using the R statistical package [53–55]. A linear mixed-effects model with sowing dates (date) and seeding rates (density) as a fixed effect, and block and year as a random effect was fit by a reduced mixed model (no significant interaction between date and density):

$$Yield \sim date + density + (1|block) + (1|year). \tag{2}$$

Analysis of variance (ANOVA) and multiple comparisons were conducted to examine the treatment differences.

To evaluate model performance and accuracy in prediction, statistical indicators that included the coefficient of determination ($R^2$), root mean square error (RMSE), normalized root mean square error (NRMSE), and the Willmott (1981) index of agreement (d) were computed from observed and simulated variables (LAI, ABIOM, and pod yield) [56]. The RMSE is the standard deviation of residuals (i.e., the differences between observed and predicted values), showing how concentrated the predicted values are around the best fit. RMSE is calculated using Equation (3).

$$RMSE = \sqrt{\frac{\sum_{i=1}^{n}(Pi - Oi)^2}{n}} \tag{3}$$

where $n$ is the number of observations, $Pi$ and $Oi$ refer to predicted and observed values, respectively. NRMSE is calculated using Equation (4).

$$NRMSE = \frac{RMSE}{M} \tag{4}$$

where $M$ is the mean of the observed variable. The index of agreement (d) is computed using Equation (5).

$$d = 1 - \left[\frac{\sum_{i=1}^{n}(Pi - Oi)^2}{\sum_{i=1}^{n}(|P'i| + |O'i|)^2}\right] \tag{5}$$

where $P'i = Pi - M$ and $O'i = Oi - M$. The d value is a better indicator of model performance, particularly relative to 1:1 line and values closer to 1 indicate better agreement between the two variables while a d value of zero indicates no predictability. Percent bias (PBIAS) measures the average tendency of the simulated data to be greater or less than the measured data. PBIAS is calculated by Equation (6).

$$PBIAS = \left[\frac{\sum_{i=1}^{n}(Oi - Pi)}{\sum_{i=1}^{n}Oi}\right] \times 100 \tag{6}$$

The optimal value of PBIAS is zero. A value of low magnitude indicates accurate model simulation. Positive values indicate model underestimation bias, and negative values indicate model overestimation bias. In addition, the model performance was assessed using the means and standard deviations of measured and predicted values.

## 3. Results

### 3.1. Sowing Date and Plant Density Effects on Yield

ANOVA results show that sowing dates had no significant effect ($p = 0.08$) on pod yield, but seeding rates had a significant effect ($p < 0.001$). Multiple comparison results show that D2 and D3 significantly increased the pod yield by 11.0% and 13.3% compared to D1, but no difference was observed between D2 and D3, although the pod yield under D3 is 2.1% higher than that under D2.

*3.2. Model Calibration*

Table 3 presents calibrated crop parameters in APEX, which were used to conduct further validation. Default values were kept for WA, HI, and DMLA because of the good fit between simulated and measured biomass, seed yield, and LAI. To increase the response of LAI to sowing density, plant population for crops and grass-1st point on curve (PPLP1) and plant population for crops and grass-2nd point on curve (PPLP2), the first and second of two points on the plant population curve were adjusted to 10.55 and 40.95, respectively. Then, the fraction of growing season when leaf area declines (DLAI), first and second point on optimal leaf area development curve (DLAP1 and DLAP2) were adjusted to fit the LAI curve. Leaf area index decline rate parameter (RLAD) and biomass–energy ratio decline rate parameter (RBMD) were adjusted to fit the ABIOM curve. Potential heat units (PHUs) for peanut were adjusted until the heat unit index (HUI, the fraction of total base-zero heat units at which operation takes place) at harvest closed to 1.0. The PHUs were set at 2000 to 2050 degree-days for different sowing dates. These above steps were repeated iteratively until reasonably accurate results were achieved.

APEX uses soil moisture content and soil temperature to estimate the timing of crop emergence. The simulated emergence was 14, 11, and 9 days for T1, T2, and T3 after sowing, as compared to the measured values of 13, 10, and 9 days (Table 2), indicating a good predictive capability. Simulated values for pod yield, LAI, and ABIOM were in good accordance with those measured in the calibration dataset (Table 4). Simulated pod yield was close to measured pod yield for calibration ($4.96 \pm 0.13$ t ha$^{-1}$ versus $4.96 \pm 0.32$ t ha$^{-1}$, Figure 4a). The performance statistics achieved in this study on pod yield ($R^2 = 0.71$, PBIAS = 0.04%) compare well with the criterion for judging satisfactory model performance ($R^2 \geq 0.60$, PBIAS < 25%) suggested by Wang et al. (2012) [49]. The model performed very well in LAI projection with a high value of $R^2$ (0.95) and d (0.98) and low PBIAS (−1.67%) and NRMSE (15.33%). The trend line of simulated LAI conformed well to peanut growth, suggesting the calibration result for LAI growth curve was acceptable (Figure 5). APEX performed well with ABIOM simulation during early growth stages though errors increased after establishing full canopy (Figure 6). This may be caused by the out-of-sync of leaf senescence and defoliation. Performance statistics for ABIOM indicated the model performance was satisfactory ($R^2 = 0.83$, d = 0.95, PBIAS = 8.53%, and NRMSE = 25.66%).

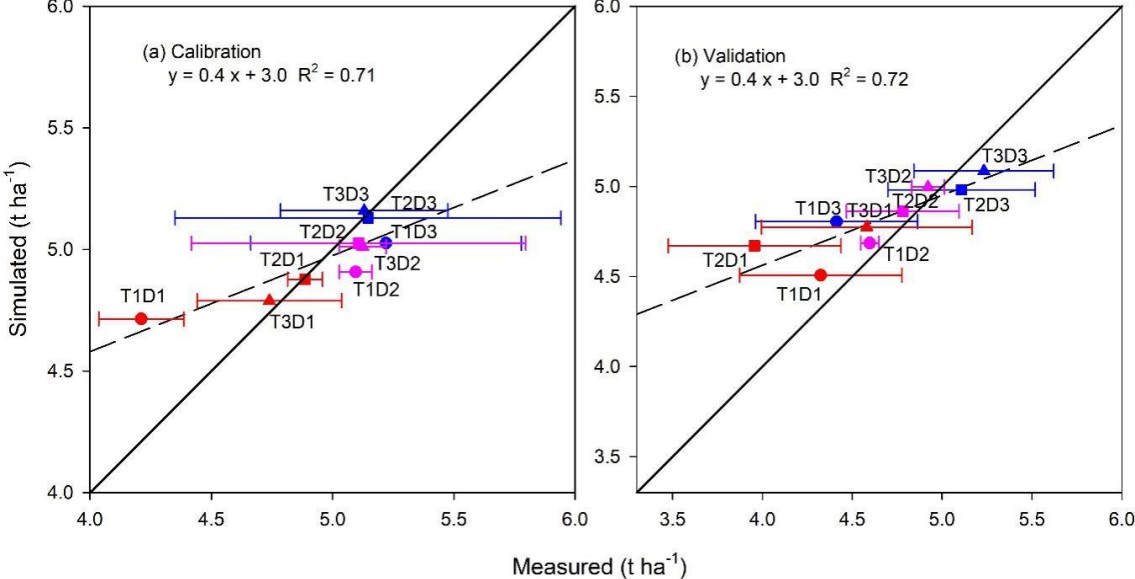

**Figure 4.** Simulated and measured pod yield of peanut under three sowing dates * three planting densities during calibration ((**a**), 2017) and validation ((**b**), 2018) stage. Values are means of three replicates. Broken line, regression line; solid line, 1:1 line. ○ represents T1, □ represents T2, △ represents T3; red represents D1, pink represents D2, blue represents D3.

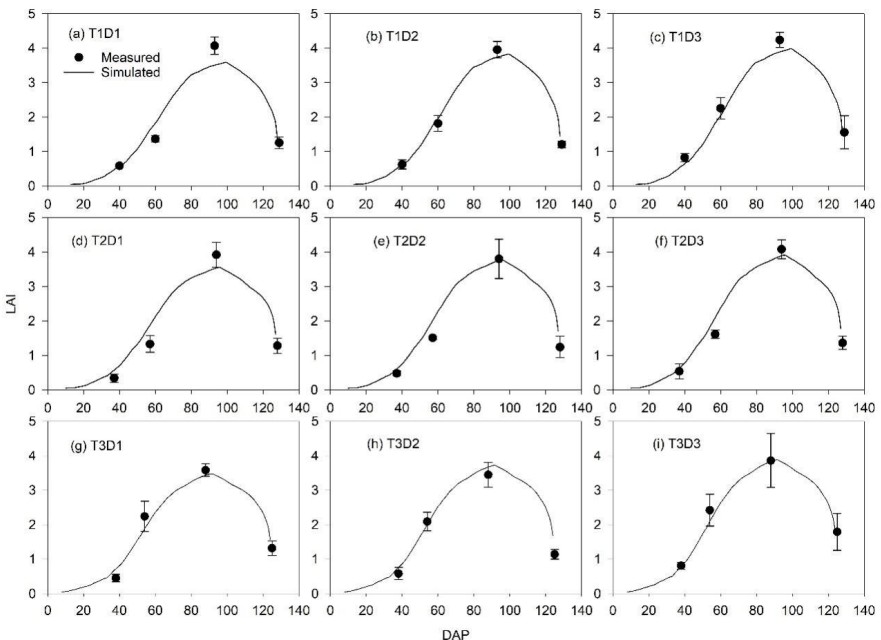

**Figure 5.** Simulated and measured leaf area index (LAI) of peanut under (**a**) T1D1, sowing on 25 April with 24 plants m$^{-2}$, (**b**) T1D2, sowing on 25 April with 30 plants m$^{-2}$, (**c**) T1D3, sowing on 25 April with 36 plants m$^{-2}$, (**d**) T2D1, sowing on 5 May with 24 plants m$^{-2}$, (**e**) T1D1, sowing on 5 May with 30 plants m$^{-2}$, (**f**) T1D1, sowing on 5 May with 36 plants m$^{-2}$, (**g**) T1D1, sowing on 15 May with 24 plants m$^{-2}$, (**h**) T1D1, sowing on 15 May with 30 plants m$^{-2}$, (**i**) T1D1, sowing on 15 May with 36 plants m$^{-2}$ during 2017. Values are means of three replicates. DAP is the number of days after planting.

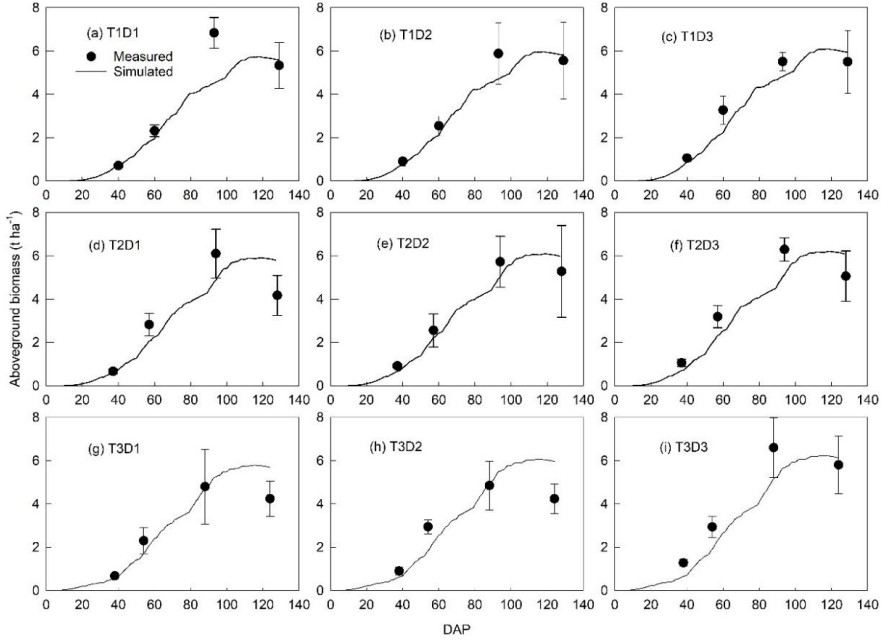

**Figure 6.** Simulated and measured aboveground biomass of peanut under (**a**) T1D1, sowing on 25 April with 24 plants m$^{-2}$, (**b**) T1D2, sowing on 25 April with 30 plants m$^{-2}$, (**c**) T1D3, sowing on 25 April with 36 plants m$^{-2}$, (**d**) T2D1, sowing on 5 May with 24 plants m$^{-2}$, (**e**) T1D1, sowing on 5 May with 30 plants m$^{-2}$, (**f**) T1D1, sowing on 5 May with 36 plants m$^{-2}$, (**g**) T1D1, sowing on 15 May with 24 plants m$^{-2}$, (**h**) T1D1, sowing on 15 May with 30 plants m$^{-2}$, (**i**) T1D1, sowing on 15 May with 36 plants m$^{-2}$ during 2017. Values are means of three replicates. DAP is the number of days after planting.

**Table 4.** Summary statistics of calibration and validation results for pod yield, leaf area index (LAI), and above-ground plant material (ABIOM).

| Stage | Index | $n$ | $R^2$ | PBIAS | $d$ | RMSE | NRMSE (%) |
|---|---|---|---|---|---|---|---|
| | Yield | 9 | 0.71 | 0.04 | 0.79 | 0.20 | 3.97 |
| Calibration | LAI | 36 | 0.95 | −1.67 | 0.98 | 0.29 | 15.33 |
| | ABIOM | 36 | 0.83 | 8.53 | 0.95 | 0.93 | 25.66 |
| | Yield | 9 | 0.72 | −3.47 | 0.82 | 0.3 | 6.37 |
| Validation | LAI | 36 | 0.86 | 7.95 | 0.95 | 0.51 | 21.69 |
| | ABIOM | 36 | 0.90 | −1.21 | 0.97 | 0.82 | 18.11 |

Note: $n$ is the number of data-points. Unit of RMSE for Yield and ABIOM is t ha$^{-1}$.

### 3.3. Model Validation

The APEX model for peanut was validated using data from the same treatments for the following year (2018) after calibration. Unlike the 2017 crop year, pest damage was reported during the 2018 crop year at these field trials. Thus, the APEX setting was modified to allow for simulating pest damage. Pest (insects and disease) factor (PST) was set to 0.85 based on field investigation results. The calibrated parameters along with the pest damage simulation yielded a good agreement between measured and simulated pod yields (4.66 ± 0.4 t ha$^{-1}$ versus 4.82 ± 0.18 t ha$^{-1}$, Figure 4b) with R$^2$ of 0.72, PBIAS of −3.47% and d of 0.82 (Table 4). APEX performed better on LAI during the calibration period than the validation (Figure 7). The model performed well in ABIOM projection with a high value of R$^2$ (0.90) and d (0.97) and a relatively low PBIAS (−1.21%) and NRMSE (18.11%) (Table 4). However, the model overestimated ABIOM during the late portion of the growing season in all T3 (mid-May sowing) scenarios (Figure 8g–i).

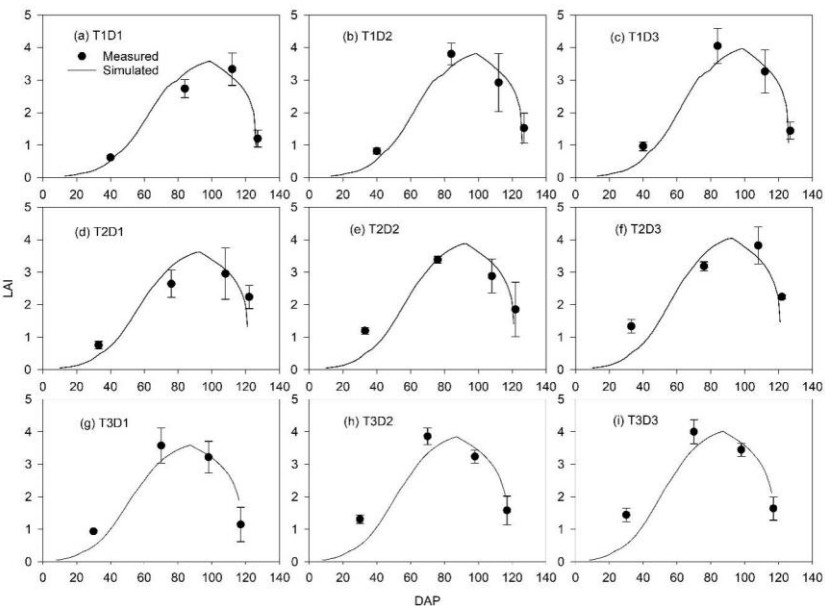

**Figure 7.** Simulated and measured LAI of peanut under (**a**) T1D1, sowing on 25 April with 24 plants m$^{-2}$, (**b**) T1D2, sowing on 25 April with 30 plants m$^{-2}$, (**c**) T1D3, sowing on 25 April with 36 plants m$^{-2}$, (**d**) T2D1, sowing on 5 May with 24 plants m$^{-2}$, (**e**) T1D1, sowing on 5 May with 30 plants m$^{-2}$, (**f**) T1D1, sowing on 5 May with 36 plants m$^{-2}$, (**g**) T1D1, sowing on 15 May with 24 plants m$^{-2}$, (**h**) T1D1, sowing on 15 May with 30 plants m$^{-2}$, (**i**) T1D1, sowing on 15 May with 36 plants m$^{-2}$ during 2018. Values are means of three replicates. DAP is the number of days after planting.

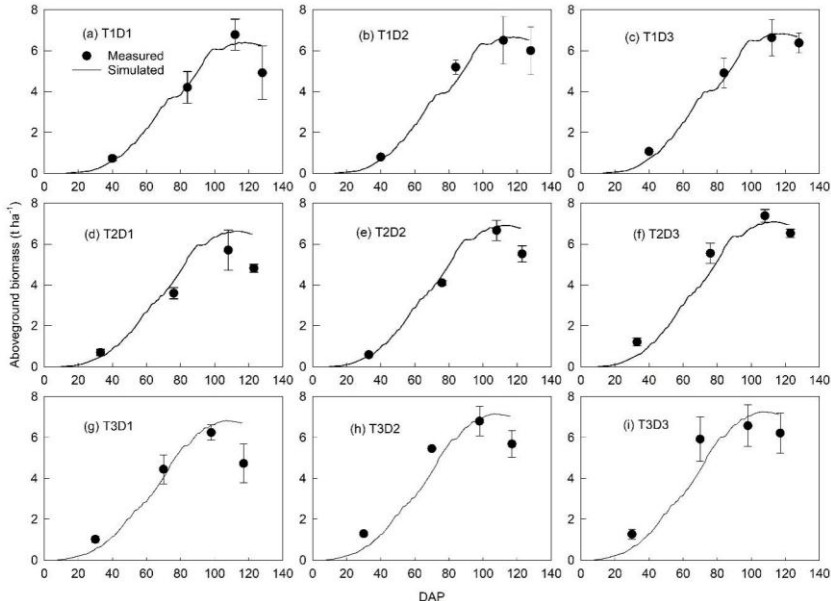

**Figure 8.** Simulated and measured aboveground biomass of peanut under (**a**) T1D1, sowing on 25 April with 24 plants m$^{-2}$, (**b**) T1D2, sowing on 25 April with 30 plants m$^{-2}$, (**c**) T1D3, sowing on 25 April with 36 plants m$^{-2}$, (**d**) T2D1, sowing on 5 May with 24 plants m$^{-2}$, (**e**) T1D1, sowing on 5 May with 30 plants m$^{-2}$, (**f**) T1D1, sowing on 5 May with 36 plants m$^{-2}$, (**g**) T1D1, sowing on 15 May with 24 plants m$^{-2}$, (**h**) T1D1, sowing on 15 May with 30 plants m$^{-2}$, (**i**) T1D1, sowing on 15 May with 36 plants m$^{-2}$ during 2018. Values are means of three replicates. DAP is the number of days after planting.

### 3.4. Optimum Seeding Rate and Sowing Rate

Pod yield and the maximum LAI under different sowing dates with rainfed irrigation were, on average, 22.8% and 15.4% lower than that with the full irrigation (Figure 9). The low pod yield (3.64 t ha$^{-1}$) and maximum LAI (2.73) at 15 plants m$^{-2}$ sharply increased in scenarios which have higher seeding rates up to 40 plants m$^{-2}$ were evaluated. Thereafter, the pod yield and maximum LAI stagnated under rainfed conditions (Figure 9a). A stronger response of pod yield and maximum LAI to seeding rates were found under full irrigation scenarios. Pod yield of peanut increased rapidly from 4.67 t ha$^{-1}$ at 15 plants m$^{-2}$ to 5.90 t ha$^{-1}$ at 40 plants m$^{-2}$. Similarly, maximum LAI increased rapidly from 3.17 at 15 plants m$^{-2}$ to 4.3 at 40 plants m$^{-2}$. Thereafter, no increase in pod yield or maximum LAI was observed under an increased seeding rate with full irrigation (Figure 9b). A greater variability was observed in pod yield and maximum LAI of all sowing dates under rainfed irrigation than full irrigation scenarios (Figure 10). The smallest pod yield (3.73 t ha$^{-1}$) of peanut sown under rainfed on 5 April sharply increased with the postponement of planting time to attain a peak (4.67 t ha$^{-1}$) on 20 May (Figure 10a). From 25 May onward, pod yields started to decrease. In contrast, with the delay of planting time, maximum LAI of peanut under no irrigation firstly decreased and then increased, and minimized LAI (3.32) was observed on 5 May (Figure 10a). Pod yields of peanut under full irrigation increased firstly and then decreased as sowing date was delayed further into late May. The highest pod yield was 5.87 t ha$^{-1}$ when peanut was sown on 10 May (Figure 10b). From 15 May onward, pod yields declined sharply as sowing dates were further delayed. Maximum LAI of peanut under full irrigation gradually declined as the planting date was delayed, ranging from 4.16 on 5 April to 3.83 on 4 June (Figure 10b).

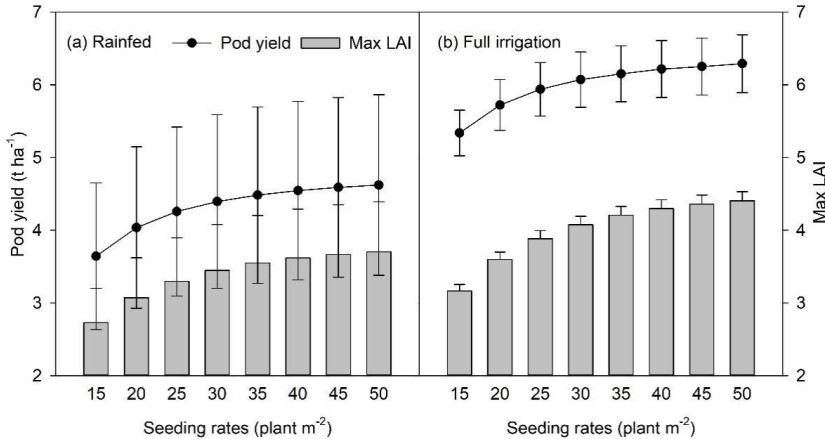

**Figure 9.** Simulated pod yield and maximum LAI of peanut of different seeding rates under rainfed (**a**) and full irrigation (**b**). Values are means of 494 data points.

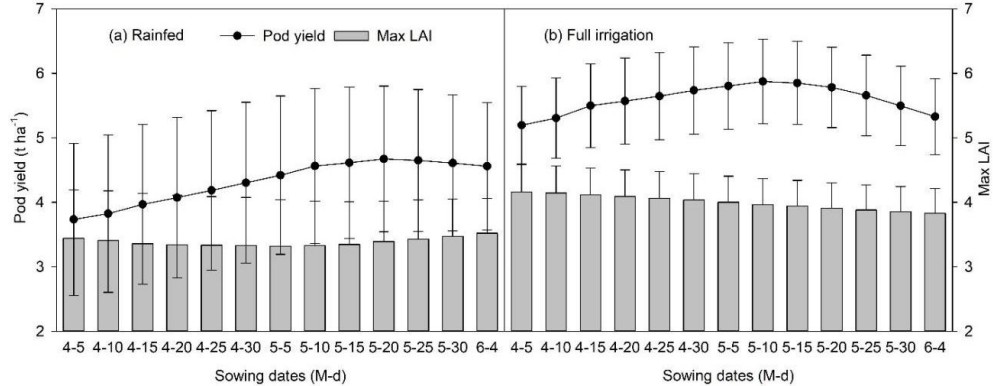

**Figure 10.** Simulated pod yield and maximum LAI of peanut of different sowing dates under rainfed (**a**) and full irrigation (**b**). Values are means of 304 data points.

### 3.5. Irrigation Scenarios, Simulation, and Analysis

Total precipitation for dry, normal, and wet years was 355 mm, 425 mm, and 510 mm, respectively. Thirty-eight simulation years were identified and categorized as 9 dry years, 19 normal years, and 10 wet years. In general, peanut pod yield, water consumption, and water use efficiency were enhanced and WS was decreased with the increase of irrigation volume and the optimum irrigation time (Figure 11). Effects of five irrigation regimes on pod yield under no other stress in different precipitation years are shown in Figure 11a. The simulated pod yields broadly ranged from 2.06 t ha$^{-1}$ in a dry year (250 mm) under rainfed condition to 7.03 t ha$^{-1}$ in a normal year (443 mm) under W21 treatment. The simulated peanut pod yield with rainfed irrigation was on average 4.80 t ha$^{-1}$ across the 38 years. The irrigation volume of 60 mm applied at sowing or flowering increased the pod yield by 14%, compared with the rainfed peanut. However, the yield increases declined to 7% when the irrigation volume increased from 60 mm to 120 mm. The simulated pod yield for peanut with W12 was on average 9% higher compared with that with W11 in dry years, but no yield increases in normal and wet years. The same trend was found in peanut yield with W12 and W22. Greater yield increases were observed when supplemental irrigation was applied in dry years (32%) than that in normal (16%) and wet years (14%) when compared to that with no irrigation.

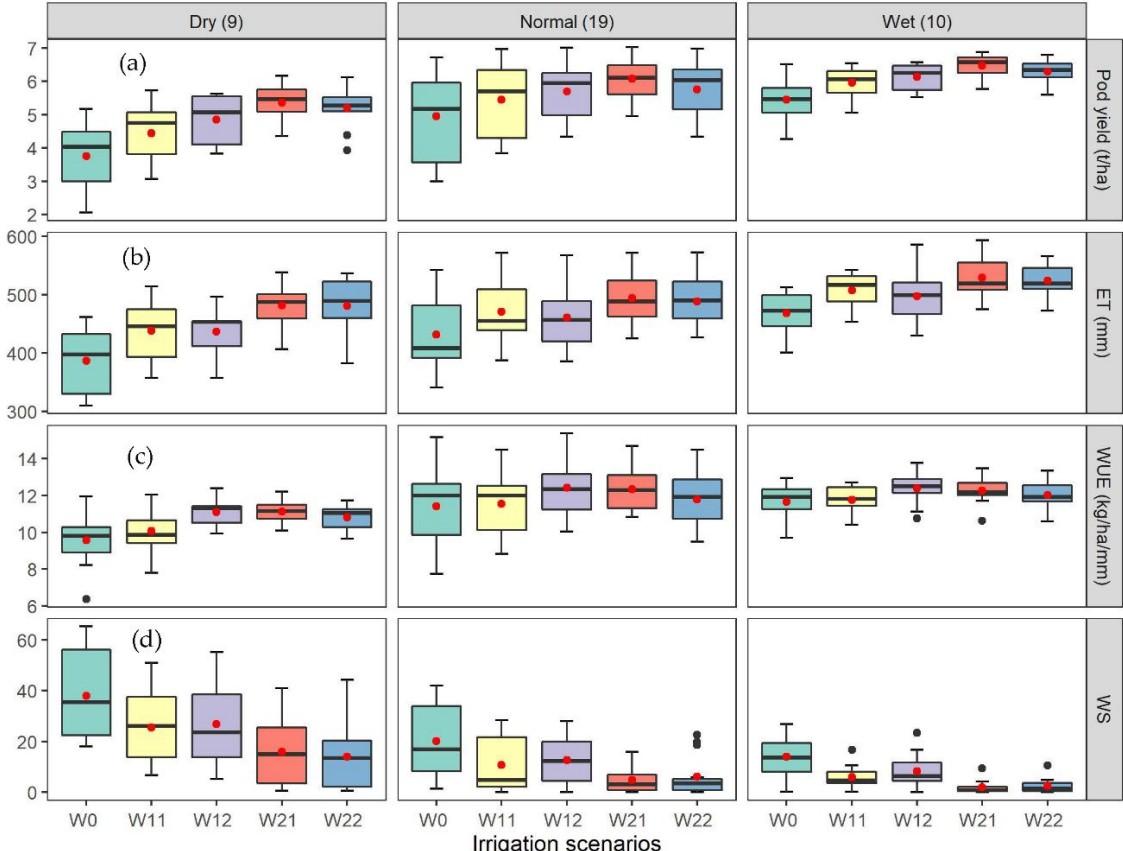

**Figure 11.** Simulated ranges of peanut (**a**) pod yield, (**b**) evapotranspiration (ET), (**c**) water use efficiency (WUE), and (**d**) water stress (WS) days under different irrigation scenarios, i.e., W0 (rainfed), W11 (60 mm depth before planting), W12 (60 mm depth at flowering), W21 (60 mm depth before planting and 60 mm depth at flowering), and W22 (60 mm depth before planting and 60 mm depth at pod set). The numbers displayed in parentheses are categorized for precipitation years. The box plots show the 25%, 50%, and 75% percentiles and the caps indicate the 5th and 95th percentiles, with the average shown by a red point. The black points are outliers.

The simulated water consumption of peanut with different irrigation regimes during 1981–2018 ranged from 310 mm in a dry year (332 mm) under rainfed conditions to 593 mm in a wet year (622 mm) under W21 treatment (Figure 11b). The water consumption of peanut was positively proportional to the irrigation rate. Similar to the case of pod yield, increased water consumption for peanut with full irrigation was observed in dry years (19%) than in normal (11%) or wet years (10%) when compared with no irrigation scenarios. The simulated water use efficiency of peanut with different irrigation regimes ranged from 6.4 kg ha$^{-1}$ mm$^{-1}$ in a dry year (250 mm) under rainfed to 15.4 kg ha$^{-1}$ mm$^{-1}$ in a normal year (425 mm) under W12 (Figure 11c). The water use efficiency of peanut increased with an increase in irrigation rate in dry years, but not in normal or wet years. Specifically, the simulated WUE was on average 5% higher with W11, and 16% higher with W12, than rainfed peanut in dry years. In that case, any further increase in the irrigation water supply will have no effect. As expected, the simulated pod yield of peanut was responsive to the WS during the growing season. The simulated WS of peanut with different irrigation regimes ranging from 65 d in a dry year (332 mm) under rainfed conditions to 0 d in wet years with irrigation (Figure 11d). The mean WS among five irrigation regimes was 24 d in dry category years, which accounted for about 19% of total peanut growing days. The normal and the wet category years had WS values of 11 d and 7 d, respectively. The WS of peanut decreased from 23 d to 7 d with an increase in irrigation rate in all precipitation categories. Impact of irrigation on WS was greater for the dry year category (24 d) than that for the normal (11 d) or the wet years (7 d) when compared with no irrigation. Overall, under the optimal sowing dates and plant

density, irrigation of 60 mm during flowering (W21) led to acceptable pod yield (5.6 t ha$^{-1}$) and ET (464 mm), which resulted in the highest water use efficiency (12.1 kg ha$^{-1}$ mm$^{-1}$).

## 4. Discussion

### 4.1. Model Calibration and Validation

The days after planting (DAP) to the emergence of peanut was positively associated with the mean air temperature [57]. We found that the DAP to emergence reduced from 13 d for an early sowing date (T1) to 7 d for a late date (T3) with the rise of air temperature. The APEX model reasonably replicated the trend of emergence date in all three sowing dates. The model successfully replicated pod yields in both calibration and validation periods as evidenced by R$^2$ values being greater than 0.7, though the pod yield was slightly underestimated during the calibration phase and slightly overestimated during the validation phase (Table 4). The low slope of yield during calibration and validation periods could be due to the moderate response of yield to seeding rate in APEX. Crop parameters PPLP1 and PPLP2 determine the response of maximum LAI to plant population and further influence dry matter production and yield formation. However, the yield did not change much when we adjusted these two parameters to fit the LAI curve. These results indicated that the yield increases slowly as the seeding rate increases. This means that the response of simulated yield is rather moderate. In contrast, increases in seeding rates can lead to significant increases in yield under field conditions. Consequently, the model ended up overestimating low yield values and underestimating high values. The differences between predicted and observed pod yield may also be driven by potential incidences of diseases and pests. This is especially true for the 2018 growing season, which had more precipitation and higher air temperature (Figure 3). Treatments with late sowing date and high plant density had greater incidences of disease and pests under a hotter and more humid environment, resulting in more yield losses. Therefore, the model tends to overestimate treatments with late sowing date and high plant density and underestimate the rest of the treatments, corresponding to the high and low yield points, respectively (Figure 4b). The performance of APEX model on yield prediction was very good when compared with good simulation (PBIAS = 12.6%) of peanut rotated with cotton under conventionally tilled or strip-tilled conditions [38] and satisfactory yield prediction (underestimate peanut yields by 20.4%) on the Little River watersheds in Georgia [39]. APEX simulations of LAI and ABIOM under 9 treatments were fairly accurate for the two growing seasons. The model predicted the ABIOM with good accuracy until 85 and 95 DAP during 2017 and 2018, respectively. After that, however, the model consistently overpredicted the ABIOM accumulation (Figures 6 and 8). This occurred as the leaf dry-weight began to decline due to disease effects and defoliation around that time for which the model did not simulate. APEX would perform better if the simulation of disease-induced defoliation is added to the crop simulation processes. A more sophisticated pest damage function than the current one in APEX can improve the simulation accuracy of peanut growth and yield as well [58–61]. The late leaf spot function incorporated into CSM-CROPGRO-Peanut model by Singh et al. (2013) is a good example for improving the accuracy of peanut growth and development simulation [61].

### 4.2. Optimum Plant Density and Sowing Date

Increased seeding rate had a positive effect on pod yield. A satisfactory yield level can be achieved with 35–40 plants m$^{-2}$ across 13 sowing dates. This may be explained by the increased LAI found in the optimum seeding rate inducing the better distribution of light and greater light interception in the canopy as well as contributing to higher absorption of nutrients from soil [20,24]. The yield was unaffected by further increasing planting density and with less than 35 plants m$^{-2}$ it was significantly reduced. These negative effects on pod yield could be due to greater competition for light and water among plants at high plant densities or the lower efficiency of resource use at extremely low plant density [23,62]. Consistent with previous observations [25,63,64], these results indicate that optimum

plant density greatly improves the canopy structure and light interception, which enhance nutrient and water use efficiency, and finally help produce the highest pod yield.

Mid-May sowings under rainfed irrigation gave the highest pod yield across 38 years and 8 planting densities, despite the lowest maximum LAI, and yield declined progressively as sowing occurred later in the year until early June despite increasing maximum LAI. Irrigation improved the pod yield and maximum LAI across 13 sowing dates, and impacted yield and LAI response to sowing dates. The optimum sowing date under full irrigation appeared 10 days earlier than that under rainfed due to the mitigation of water deficit by irrigation. Our results suggest that the most suitable sowing period falls between early to mid-May. It would be a disadvantage to the yields if sowed earlier to later. Previous studies indicate that the optimum air temperature for vegetative growth of peanut is between 30 °C and 35 °C and the ideal temperature for reproductive growth ranges between 25 °C and 28 °C. High temperatures above 35 °C during the reproductive stage reduce dry matter production, yield component, and pod yield [65–67]. Very early sowing before 30 April did not generate any advantage for yield in NCP due to water stress for vegetative growth under rainfed conditions and high-temperature stress during the pod-filling stage. In addition, a lower temperature during the early growth stage delayed flower initiation and extended maturity [21]. Therefore, it is inappropriate to sow early under the local climate and irrigation water availability. In relation to late sowing, a reduction in solar radiation and temperature during pod-filling led to a decrease in 100-seed weight and pod yield. These findings are well supported by Haro et al. (2007) who reported that delayed sowing causes a reduction in radiation use efficiency and pod yield of peanut [68]. This negative effect can be reduced by adjusting plant populations [19]. Overall, an optimum sowing date allows peanut to be exposed to appropriate temperature regimes and receive more solar radiation as well as to receive natural rainfall required for growth during the vegetative and the reproductive growth stages, enabling the crop to fulfill its yield potential.

### 4.3. Optimum Irrigation Regime

The yield reduction under rainfed ranged from 30% in dry years to 16% in wet years due to serious water stress compared to the full-irrigation (W21). There are two potential reasons for this result. In the first instance, water deficit reduced flowering and the formation of pods due to increased soil strength promoted by dry soil surface, resulting in a significant decline in pod numbers [26,69]. Secondly, water deficit inhibited canopy expansion and reduced radiation use efficiency, leading to reduced biomass production [26]. Although the ET reduction under rainfed ranged from 20% in dry years to 11% in wet years compared to the full-irrigated condition (W21), the water use efficiency of peanut did not increase. Instead, the WUE of peanut under rainfed was 14% lower than that under W21. These results indicate that rainfed peanut helped save water by not irrigating, but resulted in lower yield and WUE. Therefore, it is not recommended to grow peanut without irrigation during whole growing seasons in the NCP.

Pod yields of peanut under W12, which only irrigated 60 mm during flowering, were 29%, 15%, and 12% higher than that under rainfed in dry, normal, and wet years, respectively. Yet, pod yields of peanut under W12 were close to that under the W21 and W22 in all precipitation years. These results are well supported by previous research that the flowering and pod filling stages are the most sensitive phenological stages to soil water deficit [70,71]. Our findings also show that supplementary irrigation during the flowering stage is more efficient than that during pod filling stage. Moreover, ET reduction under W12 ranged from 10% in dry years to 6% in wet years compared to that under well-irrigated condition (W21) and resulted in higher WUE. Based on the combined effects of pod yield, ET, and WUE, peanuts can be irrigated 60 mm during the flowering stage (W12) under water shortage conditions.

### 4.4. Limitations

The current version of APEX is not able to simulate a twin-row planting pattern in one raised bed. Thus, we simplified it to a single row in the model simulation. These two kinds of planting patterns



may have the same [62] or different [72] impacts on pod yield of peanut due to differences in canopy structure and light interception. Such a simplified representation of the planting pattern may influence the simulation accuracy of the APEX model.

*4.5. Future Research*

Accounting for pest damage is crucial for models to provide useful predictions of crop production under real conditions. Although yield losses due to pest and disease damage were simulated by the model in 2018 growing season given that a fraction of the yield remained after damage, the simulated aboveground biomass was, in general, less accurate in the late-season than early to mid-season. APEX was limited in simulating the reduction of live biomass by leaf defoliation. Therefore, an enhancement is needed in the pest damage module to better represent biomass losses associated with leaf defoliation. In addition, the peanut is a legume that has considerable economic return and N-fixation as well as the forage value. Further research is needed to investigate the possibility of replacing the conventional system of winter wheat and summer maize, with a 2-year system of winter wheat/summer maize-spring peanut at the cropping system level.

## 5. Conclusions

Spring peanut is a promising alternative crop to mitigate water scarcity and environmental pollution caused by excessive water use and N inputs in conventional cropping systems in the NCP. In the present study, we evaluated the capability of the APEX model in predicting the responses of peanut to sowing dates and seeding rate in order to optimize sowing dates, seeding rate, and irrigation regimes. The results indicate that the model performed satisfactorily in simulating pod yield of peanut. The model performed reasonably well in predicting the emergence, LAI, and aboveground biomass during the majority of the growing season, with the exception of the late portion of the growing season. In the model applications for optimizing the sowing dates and seeding rates, the plant density of 35–40 plants $m^{-2}$ and sowing dates between early May (5 May) to mid-May (15 May) are the best management practices for achieving maximum yield due to the optimized climatic environment and canopy structure in respect to the growth and yield of peanut. Under optimal sowing date and plant density, a 60 mm irrigation during the flowering period led to acceptable pod yield (5.6 t $ha^{-1}$) and ET (464 mm), which resulted in the highest water use efficiency (12.1 kg $ha^{-1}$ $mm^{-1}$). Overall, the sowing density of 35–40 plants $m^{-2}$ during early May (5 May) to mid-May (15 May) with 60 mm of irrigation water can be recommended for optimum peanut performance in water constrained areas of the NCP.

**Author Contributions:** Conceptualization, J.Z., Z.Z., Q.C., and Y.Y.; investigation, Y.J., and M.S.; writing—original draft preparation, J.Z.; writing—review and editing, J.J., M.N.M., N.N., and H.Z.; visualization, J.Z.

**Funding:** This research was funded by the National Key Research and Development Program of China, grant number 2016YFD0300205-01 and the National Natural Science Foundation of China, grant number 31671640.

**Acknowledgments:** We are grateful for the constructive comments on this manuscript from the anonymous reviewers and editors.

**Conflicts of Interest:** The authors declare no conflict of interest.

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
