# Peer review of "Agricultural Policy Environmental eXtender (APEX) Simulation of Spring Peanut Management in the North China Plain"

_agronomy, doi:10.3390/agronomy9080443_

Round 1

Reviewer 1 Report

Review of agronomy-540837: APEX Simulation of spring peanut management under water-constrained agroecoregions in the North China Plain.

The manuscript aimed to calibrate and validate APEX for spring peanut in the NCP.

The paper is well designed, analysed and presented. It is concise and written carefully with clear English.

Some comments:

-         It is not clear why 2017 was used for calibration and 2018 for validation. Is 2017 grown under favourable/unstressed conditions?

-         Parameters used for calibration and validation were only yield, LAI and Biomass. Is there any data on soil water content? Flowering date to include in the calibration and validation?

-         Unit for RMSE?

-         Add NRMSE as one of statistical parameters

-         Figure 3: From the very low slope of the two Figures, it is clear that the model overestimated low yield values and underestimated high values. This has to be discussed.

-         Comment if the model has already been calibrated/validated for peanut somewhere else.

-         It appears that the data was collected using a designed field experiment. Statistical analysis (of variance) needs to be used to see the effect of the factors on the measured parameters.

Reviewer 2 Report

General comments:
The manuscript entitled "APEX Simulation of spring peanut management under water-constrained agroecoregions in the North China Plain”reports about the results of a calibration and validation exercise of the APEX cropping systems model (version 15.1.2019) for simulating spring peanut growth, LAI development and pod yield at the Wuqiao experimental station in the Northern China Plain. The experimental factors were sowing date and planting density with one variety (Juhua 25) on a Calcaric Fluvisol. In a first step, the validated model version was applied to different scenarios of sowing date and planting density to identify the optimum sowing time and planting density for the above peanut variety. In a second step, different irrigation scenarios were testing assuming optimum planting conditions using weather time series over 38 years from a nearby climate station. In general, the study has been well developed and the different steps are reasonable and justified. The presnetaiton of the results is well structure and the language appropriate.
However, there are several issues that should be clarified by the authors:
1.     The experimental layout (arrangement of planting rows and planting beds needs a drawing) and specifically the application of fertilizers (type, amount, timing) is not specified in the manuscript. Although a refrence is given for more details (38, Jiang et al. 2017), I cannot find the reference on the website of the Journal. In addition, the title of the reference (High-yield vafriety creening in Heilonggang low plain area) does not fit to the experimental design and place that is described in the manuscript.
1.     How did the authors make sure that no nutrient limitations (nitrogen, phosphorus) in this experiment. On Calcaric soils with legumes I assume that P-deficiency could be an issue.
2.     How was the model configured with respect to nutrient limitations? This is not explained at all in the manuscript. Did the authors allow the model to simulate nitrogen or phosphorus stress? If yes, they should report the output of the model with respect to these stress factors
3.     The authors did not provide information about the parameters they used to configure nitrogen fixation in the model. Did they use the nitrogen fixation routine in APEX?
4.     The authors should provide information about the application of fungizides and pesticides in the two experimental years. It is reported that in 2018, the pest damage factor (PST) was set to 0.75. What was the factor in 2017? Was the factor constant over all treatments or did the authors observe differences in the pest damage between treatments? Was the pest a disease (fungus)?
5.     Which value of PST was used in the scenario simulations?

All these issues should be adequately addressed by the authors. I recommend therefore to reconsider the paper after major revisions.

Other specific comments:
Line 1:  The title is not appropriate, because the study is not related to several agroecoregions, but only to one agro-ecoregion in China
Line 264-266: I understand that the authors used only one variety (Juhua25). Is Juhua 25 of a determinate or in-determinate growth type? How sensitive is the cultivar to photperiod? The potential heat unit requirement is a genetic property of a cultivar and should not be changed, in particular in a photoperiod-insensitive crop like peanut. Although it is reported that number of flowers and pods of certain peanut cultivars may depend on photoperiod, the timing of flowering (and maturity) was not affected. Please specify in a table how you adjusted PHU in the different treatments
Table 3: I guess that the authors are aware that the parameter HI is the maximum HI set by the user to 0.40. However, when calculating HI from the observed  data presented here, there were some treatments with a higher HI (up to 0.45). Furthermore, how did the authors set the minimum HI (WSYT) due to water stress?
Figure 3: please provide error bars of the pod yield measurements and indicate which point belongs to which treatment
Figure 10: please add the explanation for the irrigation codes in the caption of this figure. Figures should always be self-explanatory
Lines 429 ff: I understand that your peanut cultivar may have different optimum temperature depending on the development period. However, I as far as I know, APEX has only one temperature stress function for the whole growing period of a crop. Please check with the APEX documentation or the developers
Line 433: “under rainfed conditions” should be inserted after “vegetative growth”
Line 467-471: I like that you point out the limitations of the model.
Line 475: please insert “and disease” after “pest”. I assume that major damage risk is from diseases (fungi)
Line 489-491: The reader can only judge this statement when you indicate in Figure 3 which point belongs to which treatment (see comment above)
